# Development of Reliable, High Performance WLCSP for BSI CMOS Image Sensor for Automotive Application

**DOI:** 10.3390/s20154077

**Published:** 2020-07-22

**Authors:** Tianshen Zhou, Shuying Ma, Daquan Yu, Ming Li, Tao Hang

**Affiliations:** 1School of Materials Science and Engineering, Shanghai Jiao Tong University, Shanghai 200240, China; zts987@sjtu.edu.cn (T.Z.); mingli90@sjtu.edu.cn (M.L.); hangtao@sjtu.edu.cn (T.H.); 2Huatian Technology (Kunshan) Electronics Co., Ltd., Kunshan 215300, China; shuying.ma_ks@ht-tech.com; 3School of Electronic Science and Engineering, Xiamen University, Xiamen 361005, China

**Keywords:** CMOS image sensor, automotive, WLCSP, TSV, AEC-Q100

## Abstract

To meet the urgent market demand for small package size and high reliability performance for automotive CMOS image sensor (CIS) application, wafer level chip scale packaging (WLCSP) technology using through silicon vias (TSV) needs to be developed to replace current chip on board (COB) packages. In this paper, a WLCSP with the size of 5.82 mm × 5.22 mm and thickness of 850 μm was developed for the backside illumination (BSI) CIS chip using a 65 nm node with a size of 5.8 mm × 5.2 mm. The packaged product has 1392 × 976 pixels and a resolution of up to 60 frames per second with more than 120 dB dynamic range. The structure of the 3D package was designed and the key fabrication processes on a 12” inch wafer were investigated. More than 98% yield and excellent optical performance of the CIS package was achieved after process optimization. The final packages were qualified by AEC-Q100 Grade 2.

## 1. Introduction

With the innovation of commuting modes, the demand for the automotive electronic market is increasing explosively. Cars are no longer the mechanical modes of transportation but also entertainment and information centers connected with the external world by integration with advanced driver assistance systems (ADAS) [1,2,3,4]. Recently, advanced ADAS offers features of adaptive cruise control, self-parking, semi-autonomous navigation and is developing to full self-driving. As shown in Figure 1, ADAS relies on multiple data source inputs from imaging systems, such as LiDAR, radar, computer vision and in-car networking, among others [5,6,7,8].

The vision system, which is the key of the ADAS and enables the system for automatic emergency braking, autonomous driving, video mirror, rear view, 360-degree surround view and driver monitoring, benefits from the significant technological advancements of CMOS image sensor CIS [9,10,11,12]. Compared with the charge coupled device (CCD) image sensor, CIS has advantages of high integration, low power consumption and fast IO readout speed [13,14,15]. With the development of CIS, it has become one of the most widely used sensors and gradually occupies more than 99% of the share in consumer electronics market [16,17].

In recent years, CIS market size has been expanding rapidly and is expected to increase from USD 14.2 billion in 2017 to USD 24.8 billion in 2022 at a compound annual growth rate of 9.7% [18]. The main factors driving CIS market growth are trends of multi-cameras in smartphones, automotive applications, and the demand of the security and surveillance market. Among them, the automotive market is growing fastest and is forecasted to increase from USD 4.4 billion in 2018 to USD 8.7 billion in 2024, according to Yole’s report.

To meet the urgent market demand of small package size and high reliability performance for automotive CIS application, wafer level chip scale packaging (WLCSP) technology, using through silicon vias (TSV), needs to be developed to replace current chip on board (COB) packages [19,20,21,22]. Three-dimensional WLCSP with TSV technology takes advantages in shorter electrical interconnection, small form factor, high yield, and low cost [23,24,25,26,27,28]. The main challenges for CIS-WLCSP are critical reliability requirements and complex wafer level processes [29]. Previous works have focused on package fabrication, rather than reliability performance. The structure they introduced suffered different failure modes in reliability tests, such as pad delamination [21] and oxide crack [22], which were unacceptable in automotive application.

In this paper, 3D WLCSP technology for the backside illumination (BSI) CIS chip using a 65 nm node was developed. The structure design evaluated by finite element simulation and key processes on a 12 inch wafer, such as wafer level bonding, TSV formation, low temperature plasma-enhanced chemical vapor deposition (PECVD), metal shielding for solving ghost image [30] and redistribution layer (RDL) fabrication, among others, were investigated. The optical performance and reliability properties for the 3D WLCSP were investigated.

## 2. Experiments

### 2.1. CIS-WLCSP Structure

In this paper, CIS with a chip size of 5.8 mm × 5.2 mm was packaged. There were 1.4 M pixels in the center area with a 2.8 μm pixel size and 56 pads around the chip. Pad side was 85 μm × 100 μm. The WLCSP structure of CIS is presented in Figure 2a.

As shown in Figure 2b, the CIS wafer was bonded with a glass wafer through cavity wall (CV) dams. Via last TSV technology was applied to establish electrical interconnection. On the chip edge, there was a slope silicon trench, under which the tapered TSVs were fabricated in pad positions. The silicon dioxide and photosensitive polymer were used as insulation layers. Between two isolating layers, a metal shielding layer, called the IR block, was built on the backside of the chip center to solve the ghost image issue. Cu/Ni/Au RDL was made for signal transmission from pads to ball grid array (BGA) on the backside of chip. The whole package was protected by a solder mask layer (SMF).

The size of WLCSP was 5.82 mm × 5.22 mm with a thickness of 850 μm, including 400 μm AR glass and 150 μm silicon chip. The BGAs were 300 μm in diameter and 250 μm in height, respectively. The ratio of silicon-to-package was 99.27% for the WLCSP.

### 2.2. Development of Process Flow

The process flow of the CIS-WLCSP structure is shown in Figure 3. Firstly, a 40 μm height CV dam was built in the corresponding non-sensor area of the chip on a 12-inch anti-reflection glass wafer by photolithography, depicted by the “Cavity wall” in Figure 3. The material used in the CV process is the same with SMF. Then, the glass wafer was bonded with CIS wafer permanently through polymer glue by a wafer to wafer bonding process, depicted by “Bond” in Figure 3. After bonding, the CIS wafer was grinded mechanically to 160 μm and then thinned to 150 μm by dry etching for stress release, depicted by “Grind & E1” in Figure 3. The trenches-tapered TSV structures are formed by two steps of lithography (Litho trench, LE and Litho via, LV) and isotropic reactive ion etching (Etch 2, E2 and Etch 3, E3). The silicon trenches upon the chip scribe lines were etched firstly, shown by “LE & E2” in Figure 3, and following, tapered TSVs were fabricated on the positions of chip pads, shown by “LV & E3” in Figure 3. The silicon dioxide was deposited by PECVD as the first insulation layer, depicted by “PECVD” in Figure 3. For the strict temperature control of the CIS product, the deposition temperature was set to 180 °C. To solve the ghost image issue, a layer of 1.5 μm Al was sputtered on the wafer as a metal shielding layer called the IR block. Lithography and wet etching were used to remove redundant Al, shown by “IR block” in Figure 3. Next, the second passivation layer (PA) was made by spray coating only the developed areas of the via bottom where they are later exposed for oxide dry etch (OE) for electrical connection, shown by “PA & OE” in Figure 3.

Then, 300 nm Ti and 500 nm Cu were deposited as the barrier and seed layer by physical vapor deposition (PVD). Then, the Cu layer was thickened to 3.5 μm by electroplating. After the lithography of the RDL pattern and the wet etching of Ti/Cu, the RDL was formed. Following this process was electroless-plating of 3 μm nickel (Ni) and 100 nm gold (Au) on the Cu RDL, shown by “RDL” in Figure 3. To protect the chip sidewalls and die surface, the pre-cut and SMF process were performed in order. The under-bump metallization (UBM) opening was exposed and developed at the same time, depicted by “Pre-cut & SMF” in Figure 3. Hereafter, the BGAs were fabricated by ball placement and reflow process, shown by “BGA” in Figure 3. Finally, the wafer was separated to individual packages by diamond blade dicing, shown by “Dicing” in Figure 3.

### 2.3. Reliability Assessment

After assembly, the packages needed to be qualified by several standard reliability tests to assess the overall reliability. Since reliability tests are destructive, only a portion of the samples were used. In this study, 77 packages for each test were selected on different areas of the wafer. Prior to the tests, samples were baked, soaked, and reflowed. Such treatments, called pre-condition tests, are performed in order to expose real samples to thermo-mechanical stressful conditions equivalent to simulate board soldering process.

Different stresses, such as temperature, moisture, humidity or voltage, could accelerate corresponding failure modes. According to the industry standards, the thermal humidity bias storage (THB) test, thermal humidity storage (THS) test, temperature cycle (TC) test, high temperature storage (HTS) test and low temperature storage (LTS) test, were conducted for reliability evaluation. The test conditions referred to the JEDEC standards.

## 3. Results and Discussion

### 3.1. Finite Element Simulation

Generally, the coefficient of thermal expansion (CTE) mismatch between different materials in WLCSP structure could induce a high thermal stress. Typical failure modes, such as pad delamination and RDL crack in the TC test, are shown in Figure 4. The CTE and Young’s module of the related materials are shown in Table 1. To improve the reliability performance, the finite element simulation was conducted to evaluate different proposed structure designs.

#### 3.1.1. Two-Step Silicon Design

To lead out the I/O electrical signal, a two-step silicon structure was formed by plasma etching. The silicon bulk was divided by sloping trench with tapered vias, which directly impact the structure strength. The commercial software ABAQUS was employed to determine the stress distribution for different depth rations of trench and via. The strip symmetrical model is shown in Figure 5. From bottom to top, there were glass, CV dam, bonding epoxy, pad, PA, RDL and SMF. Their dimensions were set the same as the real structure.

In the simulation setting, the depths of the trench and via and diameter of via bottom were defined as Si_T1, Si_T2 and B, which are shown in Figure 4. The silicon thickness ratio levels (Si_T1/ Si_T2) were set as 90 μm/60 μm and 100 μm/50 μm. The opening diameter levels in via bottom was set as 45 μm, 50 μm and 55 μm. The temperature condition loading was set from 125 °C to −40 °C for TC test simulation. The reference temperature was set as 125 °C.

Figure 6 shows the S33 stress (stress along Z-axis direction vertical to the pad) distribution. The max value in the pad center is listed in Table 2. The max stresses for Si_T1/Si_T2 were 90/60 μm and for 100/50 μm they were 71.8 Mpa and 70.2 Mpa, which means the silicon ratio had little effect on S33 stress, while the S33 stress increased linearly as the opening diameter of via increased, presented in Table 2. With a 5 μm addition in size, the stress increased by about 7%. This was caused by SMF accumulation in via. The CTE of SMF was as high as 58 ppm/°C and when SMF shrunk with the temperature decrease, it would drag the pad up. Considering the simulation results and process capability, the structural parameters of Si_T1/ Si_T2 and via opening set as 100 μm/50 μm and 50 μm, respectively, were adopted in real structure design.

#### 3.1.2. Waterdrop Shape UBM Design 

The traditional UBM design is shown in Figure 4b, which caused nearby RDL crack failure in the TC test. To solve this problem, different kinds of UBM shape designs were evaluated by simulation and the design of experiment (DOE). A strip finite element model is shown in Figure 7. The θ refers to UBM shape, which was set as 0°, 45°, 90°, 120° and 135°. Symmetrical constraints were applied on the relevant model facets.

Figure 8 depicts the stress distribution of the RDL for different designs in the first cycle of the TC test. The simulation and corresponding DOE experiment results are listed in Table 3. As shown in Figure 8, the neck of UBM suffered a high von Mises stress. By increasing θ, the distance between the neck and the UBM increased, and the max von Mises stress decreased linearly. The stress of baseline 0° was between 45° and 90°. This indicated that the solder ball anchored RDL to inhibit its compatible deformation with SMF, which would lead to a high stress in the neck of UBM. The failure rate after 1000 cycles, presented in Table 3, confirmed the evaluation. There was no RDL crack in θ of 120° and 135° designs, which were adopted in the real structure design.

### 3.2. Optical Performance Optimization

#### 3.2.1. Flare Control—Stencil Print

Wafer to wafer bonding was the key process for CIS WLCSP. Prior to bonding, the CV dams were fabricated on glass wafer. Then, the CIS wafer was bonded with a glass wafer using bonding adhesive by a thermal compressing method. Traditionally, the adhesive was spread upon CV dams by rolling with a roller containing adhesive on its surface. Such a process was simple and cheap, while it is hard to control the glue amount. The adhesive can easily overflow the CV dam and spread along the dam’s sidewall, which is presented in Figure 9a. Therefore, the sidewalls refelcted incident light and caused serious flare image issues, as shown in Figure 9b, and deteriorated imaging quality, which was unacceptable for automotive application.

For flare reduction, the key is to control the amount of bonding glue. Stencil printing technology was introduced in the bonding process to solve this problem. The schematic of stencil printing is shown in Figure 10a. After the alignment of stencil and glass, bonding glue was printed into the surface of the CV dams through stencil opening. By optimizing the stencil design and process parameter, the overflow of adhesive was controlled within the width of a dam tooth (typically 50 μm), which was also designed for absorbing extra glue. Figure 10b shows the glue boundary exceeding CV dam was less than 23 μm for the outer-dam and not obvious for the inner-dam. In total, 13 areas of wafer and 5 for each were confirmed. No obvious flare issue was found for the final package module in the optical function test. 

To increase light transmittance for better image quality, double coating AR glass was applied in CIS-WLCSP for automotive products. The transmittance and reflectance spectra of the AR glass are shown in Figure 11. The transmittance of the light with a wavelength of 420~900 nm was more than 97% with 0° and 20° incident angle as presented in Figure 11a. When the angle of incident light increased to 35°, the transmittance was still as high as 96.5% for light with a wavelength of 420~850 nm. With the increase in incident angle, only light with larger wavelength, which was out of the visible light spectrum, had an obvious transmittance loss. Meanwhile, the reflectance was lower than 1.5% for light with wavelength from 400 nm to 900 nm, as shown in Figure 11b.

#### 3.2.2. Ghost Image Solution—IR Block

Compared with traditional COB packages, CIS-WLCSP was assembled based on a chip-unit without substrate, and the module was soldered on PCB directly. When infrared light interfered from the backside of the sensor module, it could penetrate the SMF, PA and Silicon into the diode area and mixed with the optical signals, resulting in ghost image issue. Such an issue was not acceptable for automotive application, based on security consideration. To solve this problem, a layer of metal Al, called an IR-block, was applied and the schematic is presented in Figure 12a. 

To form the IR-block, 1.5 μm Al was sputtered on a CIS wafer. After lithography and wet etching for redundant Al (outside the sensor area) removement, the IR-block layer was fabricated, as shown in Figure 12b.

### 3.3. Key Process Results

#### 3.3.1. Wafer Thinning—Grind and E1

After the bonding of the CIS wafer and glass, the following processes were performed on the silicon side. Firstly, the CIS wafer was thinned mechanically to 160 μm after coarse and fine grinding. Mechanically grinding would induce stress and micro-damage on the wafer surface, which could result in a chip crack in following processes. To release the mechanical stress and eliminate micro-damage, about 10 μm silicon was removed by reactive ion etching. The results of the silicon thickness measured after the “Grind” and “E1” processes are listed in Table 4. The average thickness of five areas was 158 μm in the “Grind” process and 151 μm in the “E1” process. The number of each point met the corresponding spec requirement within a variation of 5 μm.

#### 3.3.2. TSV Formation—E2 and E3

To lead out the I/O electrical signal, a two-step silicon structure was formed by isotropic reactive ion etching. The sloping trench was fabricated in the first step, and then tapered TSVs were formed in the bottom of the trench. Such a structure was widely adopted in CIS and fingerprint products with pads around the chip. Compared with vertical TSV structure, the two-step silicon structure was easier for manufacturing, especially for isolation layer formation, seed layer deposition and RDL lithography.

Since the second passivation layer was formed by a spray coating method, the shape of the silicon trench and via would impact PA thickness on the structure sidewall directly. To ensure the PA thickness for insulation, the angle of the trench was designed to 70°. After process optimization, the profile of the trench was controlled with straight and smooth form and without sharp angle and bowl shape in the top corner, as shown in Figure 13a. The top opening measured in the X direction and the depths of the trenches in five areas of the wafer are listed in Table 5. The spec for two items were 500 ± 20 μm and 100 ± 5 μm, respectively, which were achieved.

The requirements for the profile of tapered TSV were same with the silicon trench, which were also achieved and presented in Figure 13b. The top opening width, bottom opening width, and depth of the via in five areas of the wafer were listed in Table 6. The average of the top opening width, bottom opening width and the depth of the via were 86 μm, 52 μm and 51 μm, respectively, and met the corresponding spec.

#### 3.3.3. First Passivation Layer—Low Temperature PECVD

For consumer products, only one layer of passivation was used as insulation layer and generally the material applied was a polymer. While for the automotive application, the assembled module needs to pass more strict reliability test conditions, especially for those with bias. To meet higher reliability requirements, two passivation layers were developed, concluding silicon dioxide and polymer. The oxide was deposited by PECVD and the second passivation layer was formed using polymer by spraying coating. 

Before the PECVD process, it was important to remove the by-product formed in the dry etch process by proper pre-clean method. In this study, single wafer megasonic cleaning with SC1 chemical was introduced and cooperated with regular wafer clean method as IPA clean for the pre-clean of PECVD. Since the micro-lens in the CIS device could not withstand a temperature higher than 200 °C for hours, the deposition temperature in PECVD was set at 180 °C. Compared with high temperature PECVD, lower temperature means lower reactive rate and smaller step coverage.

By the optimization of the PECVD process, the thickness uniformity in bare silicon wafer was higher than 98% and the step coverage in via corner for this trench-via structure was more than 30%. The outlook of the wafer after the PECVD process is shown in Figure 14a. The SEM cross-section images shown in Figure 14b presented that the thickness of the oxide was 0.93 μm, 0.92 μm and 0.42 μm in die surface, trench sidewall and via corner.

As shown in Table 7, the average thicknesses of oxide in die surface, trench sidewall and via corner for five areas on the wafer were 0.89 μm, 0.90 μm and 0.41 μm, respectively, which met the design spec.

#### 3.3.4. Second Passivation Layer

After the first passivation layer deposition and IR-block formation, a second passivation layer was fabricated by spray coating, which not only served as an insulation layer for preventing electrical short or leakage but also acted as a compensation layer for mechanical and thermal stress buffer. To lead out the electrical signal from the chip pad, the area of via bottom need to be exposed and developed. Then, plasma etching was performed to remove the oxide upon the pad. Figure 15 presents the outlook of chip after the second passivation layer formation. The opening of PA in the via bottom was set at 25 μm.

As shown in Table 8, the average thicknesses of PA in die surface, trench sidewall and via corner for five areas in the wafer were 6.2 μm, 5.8 μm and 2.9 μm, respectively, which met the design spec.

#### 3.3.5. RDL Formation

A layer of RDL was fabricated to lead out the electrical signal from pad to backside BGAs. The RDL was formed by the following steps: seed layer deposition, copper electroplating, RDL pattern lithography, wet etch of seed layer, photoresist removement and Ni/Au electroless-plating. 

Firstly, 300 nm Ti and 500 nm Cu were deposited as the barrier and seed layer by PVD. Then electroplating was performed to increase Cu thickness to 3 μm. After lithography by spray coating, the RDL pattern was formed. Next, the redundant Cu and Ti seed layers without photoresist protection were removed by wet etching. The following process was photoresist removal and the electroless-plating of 3 μm nickel (Ni) and 100 nm gold (Au) on Cu RDL. The outlook of the wafer after electroless-plating Ni/Au is shown in Figure 16.

As shown in Table 9, the average thicknesses of Cu, Ni, and Au for five areas in the wafer were 2.5 μm, 3.2 μm and 97 nm, respectively, which met the design spec.

#### 3.3.6. SMF and BGA Formation

To protect the chip sidewalls and RDL trace, SMF was formed by spin coating. The UBM opening was exposed and developed at the same time, which is shown in Figure 17. The opening of the UBM was set as 250 μm.

After SMF formation, the BGAs were fabricated by ball placement and reflow process. The diameter of the solder ball was 300 μm and the material was SAC305. The outlook of BGA is shown in Figure 18 with corresponding SEM cross-section. The interfacial strength of the solder ball and UBM were measured by ball shear test. As shown in Table 10, the average ball height, ball diameter and ball shear strength for five areas on the wafer were 251 μm, 308 μm and 308 gf, respectively, which met the design spec.

### 3.4. Yield of CIS-WLCSP

After process optimization, three device wafers were assembled for quality verification. The outlook of CIS-WLCSP was presented in Figure 19. According to electrical and optical inspection results, the average yield for low-volume production had reached approximately 98%.

The cross-section SEM images of CIS-WLCSP were presented in Figure 20. The silicon thickness of the trench and depth of via were 96 μm and 52 μm, respectively. The opening diameter in via bottom was 47 μm. The thickness of different layers, including oxide, PA, RDL, SMF and sidewall-protection, met the design spec, respectively. 

### 3.5. Reliability Results

For reliability qualification, 77 packaged samples for each test were selected randomly from assembled wafers. Prior to the THS, THB and TC-B test, samples need to be baked at 125 °C for 24 h, soaked at 30 °C under 60% relative humidity (RH) for 192 h and reflowed three times with a 260 °C peak temperature, which were included in pre-con test.

Then the THB test at 85 °C under 85% RH, THS test at 85 °C under 85% RH, TC-B test (−55~125 °C), HTS test at 125 °C and LTS test at −40 °C were performed. The test read points and results are presented in Table 11. All samples passed the tests, and the CIS-WLCSP was qualified by AEC-Q100 Grade 2 level.

## 4. Conclusions

To meet the high reliability and optical requirements for automotive CIS application, a WLCSP with a thickness of 850 μm was developed for BSI-CIS chip using a 65 nm node with size of 5.8 mm × 5.2 mm. The mechanical structure design was evaluated by finite element simulation, which provided references for real structure fabrication. Several key processes on a 12” inch wafer, such as wafer bonding, TSV formation, low temperature PECVD, metal shielding for solving ghost image and RDL fabrication, among others, were investigated. More than 98% yield of low-volume production and excellent optical performance was achieved after process optimization. The module assembled has 1392 × 976 pixels with a frame rate up to 60 frames per second with more than a 120 dB dynamic range. The final package was qualified by AEC-Q100 Grade 2. The results demonstrated that WLCSP using TSV technology was a reliable packaging technology for automotive CIS application, which meanwhile could meet the demand of miniaturized package size, higher performance and integration density.

## Figures and Tables

**Figure 1 sensors-20-04077-f001:**
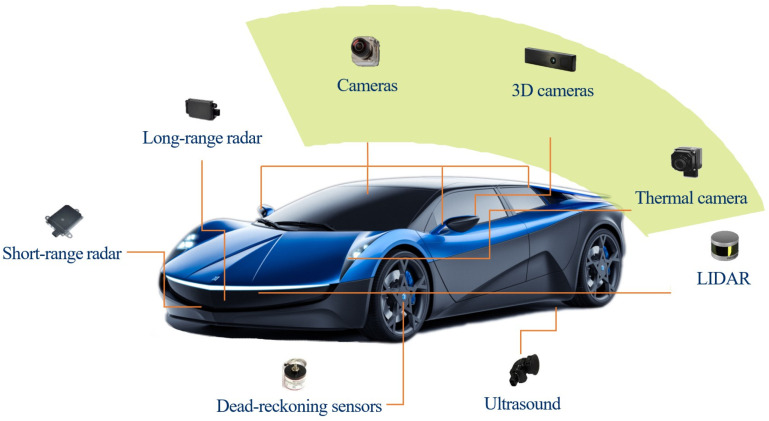
Image devices used in ADAS for automotive application.

**Figure 2 sensors-20-04077-f002:**
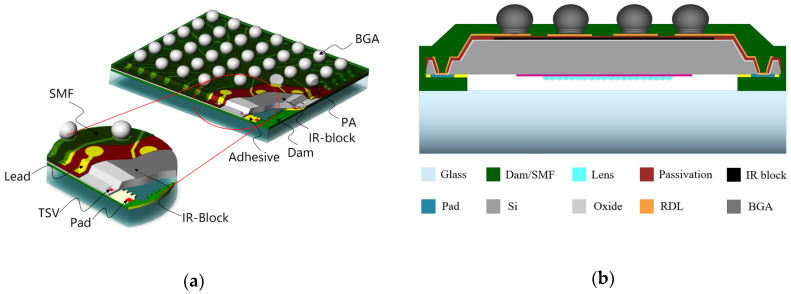
The schematic of the package (**a**) 3D view of the CIS-WLCSP structure; (**b**) Cross-section of the CIS-WLCSP structure.

**Figure 3 sensors-20-04077-f003:**
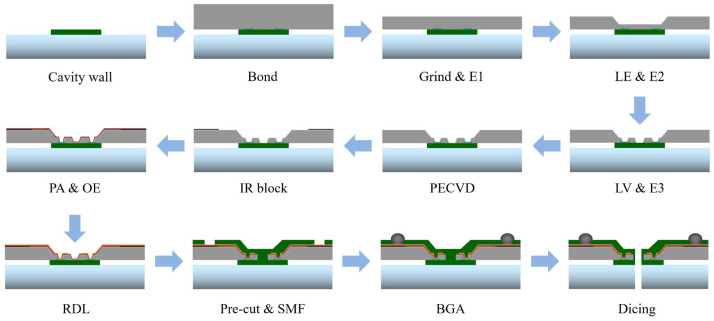
The schematic of CIS-WLCSP process flow.

**Figure 4 sensors-20-04077-f004:**
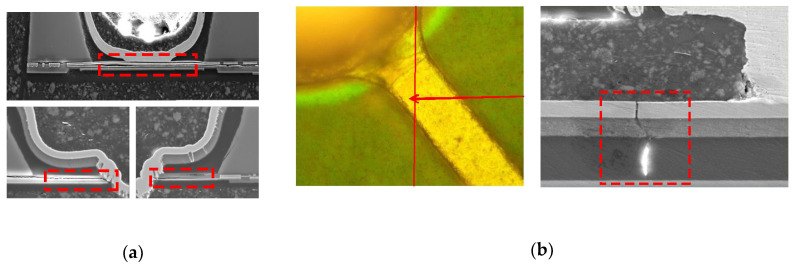
Package failure modes after TC test. (**a**) Pad delamination in the bottom of the TSV; (**b**) RDL crack near the UBM neck.

**Figure 5 sensors-20-04077-f005:**
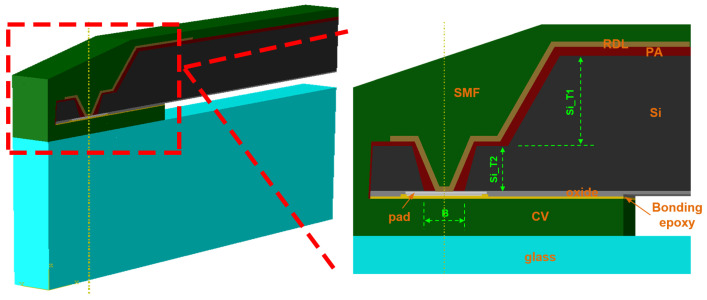
The strip model of the WLSCP in ABAQUS.

**Figure 6 sensors-20-04077-f006:**
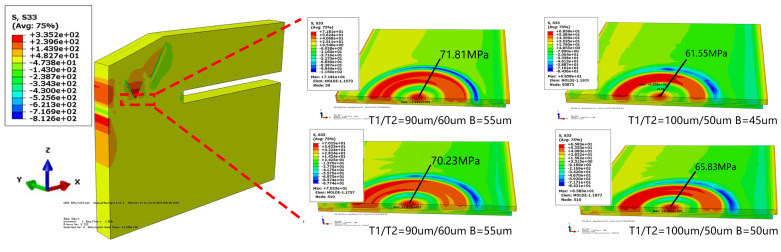
Stress distribution of pad in one TC cycle.

**Figure 7 sensors-20-04077-f007:**
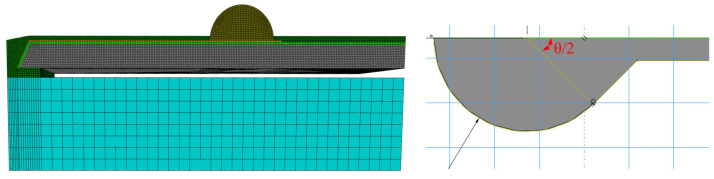
The strip model of WLCSP for different UBM design.

**Figure 8 sensors-20-04077-f008:**
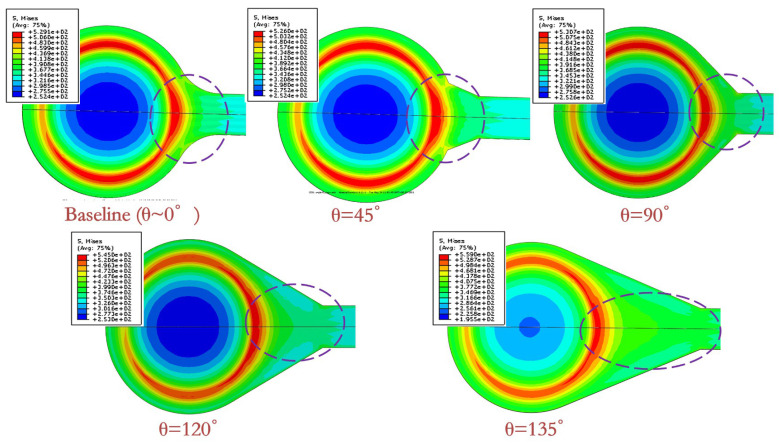
Stress distribution of pad in one TC cycle.

**Figure 9 sensors-20-04077-f009:**
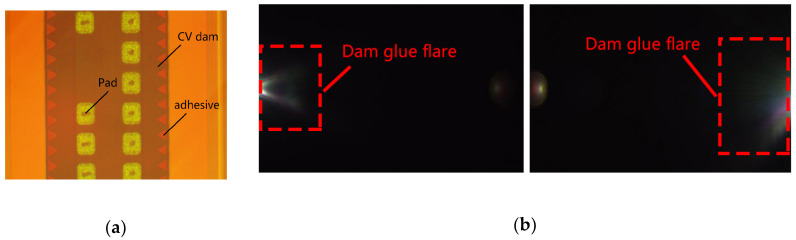
(**a**) Adhesive overflowed the CV dam; (**b**) flare image issue caused by dam glue reflection.

**Figure 10 sensors-20-04077-f010:**
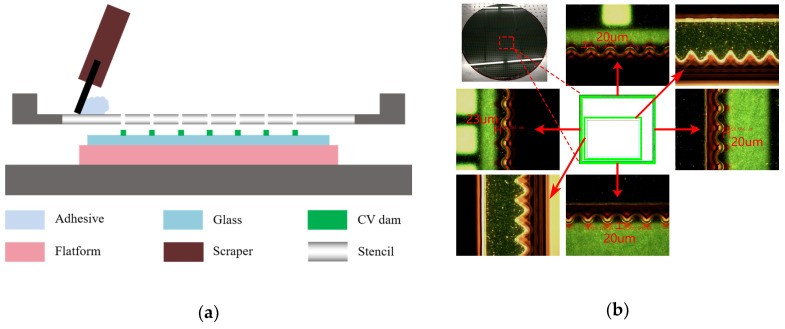
(**a**) Schematic of adhesive glue stencil print; (**b**) Adhesive glue control within CV dam tooth.

**Figure 11 sensors-20-04077-f011:**
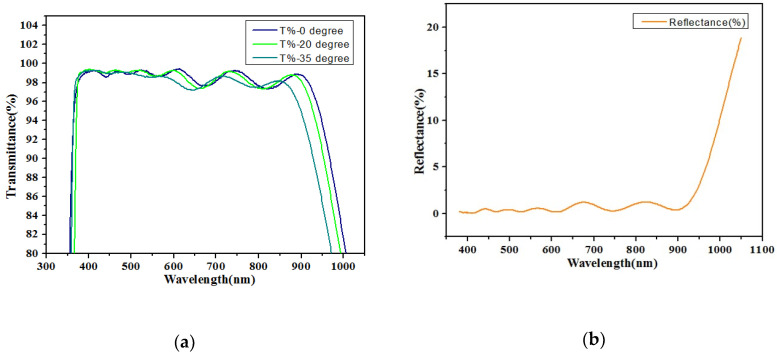
(**a**) Transmittance and (**b**) reflectance spectra of the double coating AR glass.

**Figure 12 sensors-20-04077-f012:**
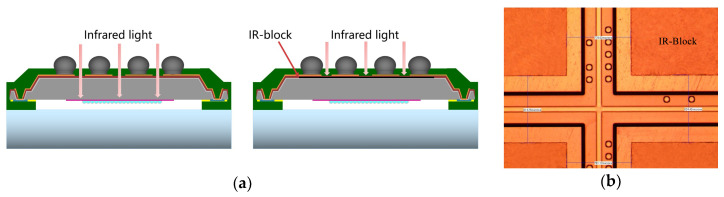
(**a**) Schematic of IR-block; (**b**) OM image of IR-block process.

**Figure 13 sensors-20-04077-f013:**
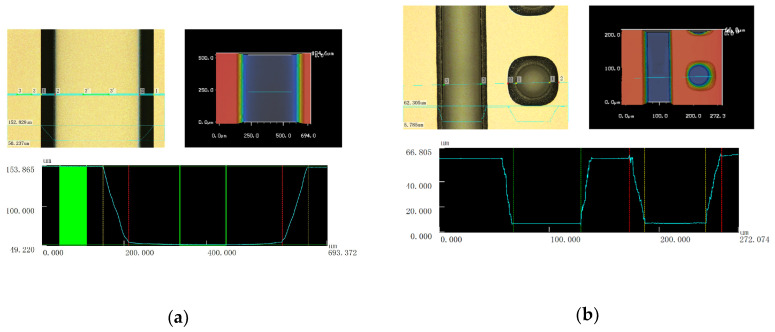
Three-dimensional microscope image of (**a**) the silicon trench and (**b**) the trench and tapered via.

**Figure 14 sensors-20-04077-f014:**
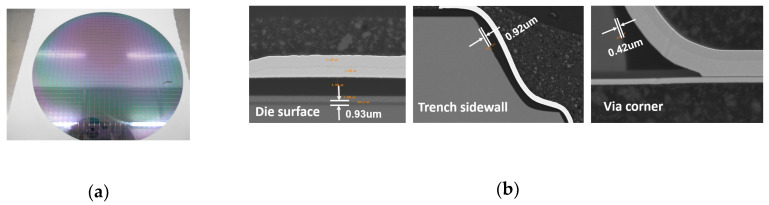
(**a**) The outlook of the wafer after PECVD process. (**b**) The SEM cross-section images of die surface, trench sidewall and via corner.

**Figure 15 sensors-20-04077-f015:**
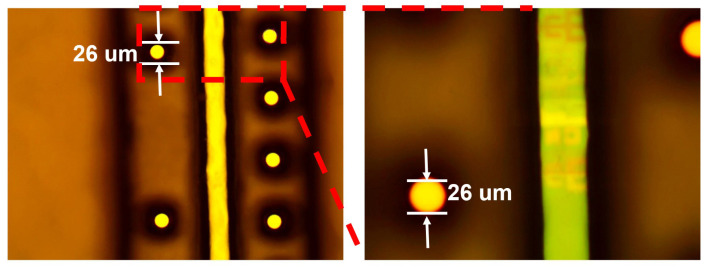
The outlook of the via after second passivation layer formation.

**Figure 16 sensors-20-04077-f016:**
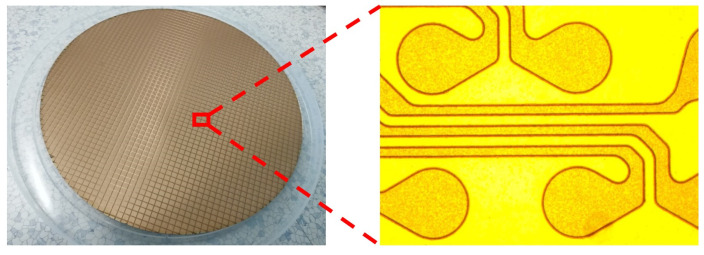
The outlook of the wafer after electroless-plating Ni/Au.

**Figure 17 sensors-20-04077-f017:**
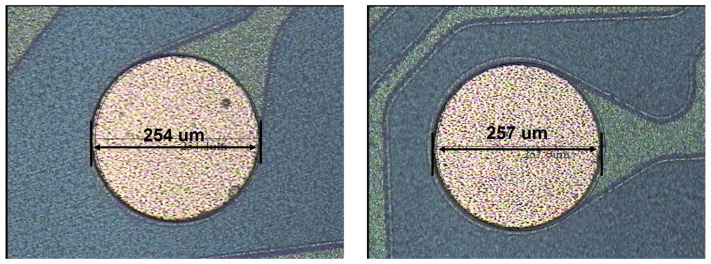
The OM image of SMF and UBM opening.

**Figure 18 sensors-20-04077-f018:**
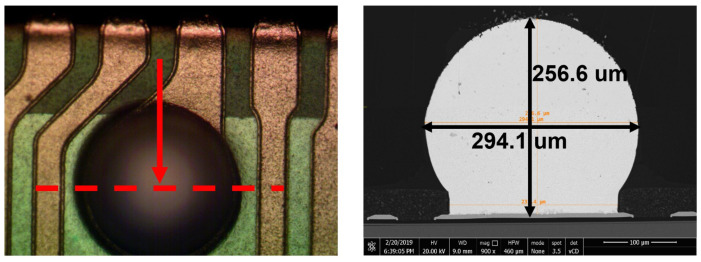
OM image and cross-section SEM image of the solder ball.

**Figure 19 sensors-20-04077-f019:**
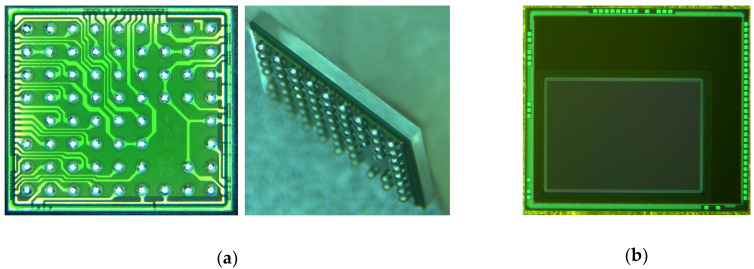
The outlook of CIS-WLCSP (**a**) BGA side; (**b**) Sensor side.

**Figure 20 sensors-20-04077-f020:**
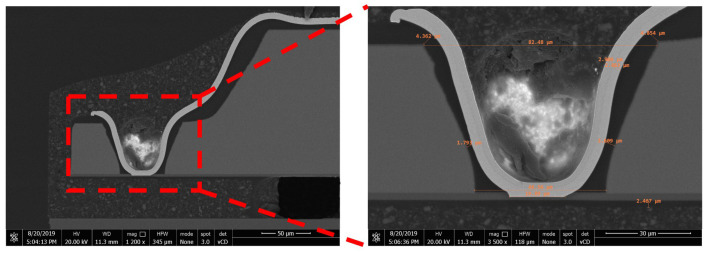
The cross-section SEM image of CIS-WLCSP.

**Table 1 sensors-20-04077-t001:** Material property related to the package.

Property/Material	Si	SiO_2_	PA	Cu	SMF
CTE (ppm/°C)	2.8	0.5	54.0	16.9	58.0
Young’s module (GPa)	131.0	70.0	2.5	129.0	4.6

**Table 2 sensors-20-04077-t002:** The max S33 stress in the pad center for different structure design.

Value/Condition	1	2	3	4
Si_T1/Si_T2 (μm)	90/60	100/50	100/50	100/50
Via opening (μm)	55	55	50	45
Max stress (MPa)	71.8	70.2	65.8	61.5

**Table 3 sensors-20-04077-t003:** The max von Mises stress of different UBM design.

Value/θ	0°	45°	90°	120°	135°
Max stress (MPa)	437.7	458.2	428.2	410.7	406.5
Failure rate of TC 1000 cycle	6.7%	8.9%	4.4%	0	0

**Table 4 sensors-20-04077-t004:** The silicon thickness in Grind and E1 process.

Process	Point 1	Point 2	Point 3	Point 4	Point 5	AVG.	SPEC
Grind/μm	157	157	158	159	157	158	160 ± 5
E1/μm	147	153	154	150	150	151	150 ± 5

**Table 5 sensors-20-04077-t005:** The top opening and depth of the trench in E2 process.

Item	Point 1	Point 2	Point 3	Point 4	Point 5	AVG.	SPEC
Top opening/μm	504	503	496	504	500	501	500 ± 20
Depth/μm	102	101	101	102	101	101	100 ± 5

**Table 6 sensors-20-04077-t006:** The top opening width, bottom opening width and depth of via in the E3 process.

Item	Point 1	Point 2	Point 3	Point 4	Point 5	AVG.	SPEC
Top opening/μm	84	84	86	87	89	86	85 ± 10
Bottom opening/μm	49	52	53	53	52	52	50 ± 5
Depth/μm	49	50	53	52	53	51	50 ± 5

**Table 7 sensors-20-04077-t007:** The thicknesses of oxide in die surface, trench sidewall and via corner in PECVD process.

Item	Point 1	Point 2	Point 3	Point 4	Point 5	AVG.	SPEC
Die surface/μm	0.87	0.88	0.87	0.90	0.93	0.89	0.9 ± 0.2
Trench sidewall/μm	1.03	0.73	1.01	0.83	0.92	0.90	>0.3
Via corner/μm	0.39	0.40	0.36	0.50	0.42	0.41	>0.3

**Table 8 sensors-20-04077-t008:** The thicknesses of PA in die surface, trench sidewall and via corner in second PA process.

Item	Point 1	Point 2	Point 3	Point 4	Point 5	AVG.	SPEC
Die surface/μm	6.1	6.3	6.1	5.8	6.5	6.2	6 ± 2
Trench sidewall/μm	6.2	5.5	5.3	5.7	6.5	5.8	>2
Via corner/μm	2.6	3.4	2.0	3.3	3.3	2.9	>2

**Table 9 sensors-20-04077-t009:** The thicknesses of Cu, Ni, and Au in RDL formation process.

Item	Point 1	Point 2	Point 3	Point 4	Point 5	AVG.	SPEC
Cu/μm	2.5	2.3	2.4	2.7	2.6	2.5	2.5 ± 1
Ni/μm	3.3	3.1	3.2	3.3	3.1	3.2	3 ± 1
Au/nm	99	98	97	96	97	97	>60

**Table 10 sensors-20-04077-t010:** The ball height, ball diameter and ball shear strength of solder ball in BGA process.

Item	Point 1	Point 2	Point 3	Point 4	Point 5	AVG.	SPEC
Height/μm	248	250	260	245	251	251	250 ± 30
Diameter/μm	307	312	310	306	306	308	300 ± 30
Shear strength /gf	330	322	291	287	309	308	>200gf

**Table 11 sensors-20-04077-t011:** Reliability test items and results.

Test Item	Condition	Read Point	Samples	Failure
Pre-con	MSL3(30 °C /60%RH)	192 h	231 ea/lot*3	0
HTS	125 °C	1008 h	77 ea/lot*3	0
LTS	−40 °C	1008 h		0
THS	85 °C/85%RH	1008 h		0
THB	85 °C/85%RH +Bias	1008 h		0
TC-B	−55 °C~125 °C	1000 cycles		0

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
