# Peer review of "Development of Reliable, High Performance WLCSP for BSI CMOS Image Sensor for Automotive Application"

_sensors, 2020, doi:10.3390/s20154077_

Round 1

Reviewer 1 Report

This paper presents comprehensive method for the WLCSP technology of image sensor for automotive application. Each processing steps are well described and simulation and test results are clearly shown.

I still have several minor comments.

Line58: since the ‘PECVD’ appears first time here in this paper, please write full name here not in the line 89.

Line 59: since the ‘RDL’ appears first time here in this paper, please write full name here not in the line 72.

Figure 1: The characters are too small and nobody read it.  Please enlarge the characters or even remove this figure since this is not so important in this paper.

Figure 2:  Please show formal name of ‘PA’ in main text.

Figure 2(a): The figure is a bit dark and difficult to distinguish each part. I recommend to enlarge the picture.

Figure 6: Please enlarge the scale and the xyz direction.

Figure 8: Please enlarge the scale.

Line 332-334: remove ‘which brought  … systems application’. This sentence is OK in introduction, but not related to the conclusion of this paper.

Author Response

Thank you very much for your comments, please see the attachment.

Reviewer 2 Report

Great work by authors in an area that is seeing immense growth. Can you also comment on the price per unit of such packaging? 

What tests were run to obtain the reliability results? Was it a connectivity check or was it an imaging test? Did you notice any image degradation due to condensed water droplets inside the chip package?

Author Response

(The authors gave the same response as above.)

Reviewer 3 Report

Review comments:

  • 2 line 64: “There were 8M pixel in center area…”
    • Do you mean 8Mpixel resolution? In the abstract a pixel resolution of 1392x976 (1.3Mpixel) is mentioned.
    • It might also be useful to mention pixel pitch
  • 2 line 74: “thickness of 750um…”
    • 400um AR glass + 150um silicon + 125um BGA = 675um: can you elaborate on the remaining 75um? Is this mostly taken by CV/cavity (in Fig 20 this is ~50um)? See also later comment on Table 10.
  • 3 line 80:
    • “The process flow of CIS-WLCSP structure was shown…”. I recommend using present tense instead of past tense throughout the text, so “The process flow of CIS-WLCSP structure is shown…” etc.
    • “Firstly, the CV dams were built…”: can you specify thickness and material (SMF?)
  • 4 line 135: ”…which were showed in Fig.4.” should be “…which are shown in Fig.4.”
  • 6 Fig. 7 & 8: From these figures, it is not clear how angle theta is defined. In Fig.7, theta seems to be close to 45 degrees, but the shape looks similar to angle 90 in Fig.8. Angle in Fig.7 is probably theta/2?
  • 7 Fig.9(a):
    • it is not clear if this picture is taken with CV dam present or after CV dam removal (during failure analysis)
    • line 176: “ …flare image issue showed…” should be “…shown in Fig. 9(b)”.
  • 8 lines 200-205: is the IR-block layer deposited to block infrared light from backside of package, or to ensure uniform reflection of infrared light originally impinging on front side (bottom in Fig 12)?
  • 11 line 288: “97mm” should be “97nm”
  • 12 lines 298-300: “average ball height….251um” on p.2 the ball height is said to be 125um in height (250um in diameter)
  • 13 line 332: “The module assembled has 1392×976 pixels with a resolution at up to 60 frames per second with more than 120 dB dynamic range…” suggested to change into “The module assembled has 1392×976 pixels with frame rate up to 60 frames per second with more than 120 dB dynamic range…”

Author Response

(The authors gave the same response as above.)
